



# Identifying mountain permafrost degradation by repeating historical ERT-measurements

Johannes Buckel[1], Jan Mudler[1], Rainer Gardeweg[2], Christian Hauck[2], Christin Hilbich[2], Regula Frauenfelder[3], Christof Kneisel[4], Sebastian Buchelt[4], Jan Henrik Blöthe[5], Andreas Hördt[1], Matthias Bücker[1]

[1]Institute for Geophysics and Extraterrestrial Physics, Technische Universität Braunschweig, Braunschweig, 38106, Germany
[2]Department of Geosciences, University of Fribourg, 1700 Fribourg, Switzerland
[3]Norwegian Geotechnical Institute, Sandakerveien 140, 0484 Oslo, Norway
[4]Department of Physical Geography, Institute of Geography and Geology, University of Würzburg, 97074 Würzburg, Germany
[5]Institute of Environmental Social Sciences and Geography, University of Freiburg, 79085 Freiburg, Germany

*Correspondence to*: Johannes Buckel (buckel@posteo.de)

**Abstract.**

Ongoing global warming intensifies the degradation of mountainous permafrost. Permafrost thawing impacts landform evolution, reduces fresh water resources, enhances the potential of natural hazards, and thus has significant socio-economic impact. Electrical resistivity tomography (ERT) has been widely used to map the ice-containing permafrost by its resistivity contrast compared to the surrounding non-frozen medium. This study aims to reveal the effects of ongoing climate warming on alpine permafrost by repeating historical ERT and analysing the temporal changes in the resistivity distribution. In order to facilitate the measurements, we introduce and discuss the employment of textile electrodes. These newly developed electrodes significantly reduce working effort, are easy to deploy on blocky surfaces, and yield sufficiently low contact resistances. We analyse permafrost evolution on three periglacial landforms (two rock glaciers and one talus slope) in the Swiss and Austrian Alps by repeating historical surveys after periods of 10, 12, and 16 years, respectively. The resistivity values have been significantly reduced in ice-poor permafrost landforms at all study sites. Interestingly, resistivity values related to ice-rich permafrost in the studied active rock glacier partly increased during the studied time period. To explain this apparently counterintuitive (in view of increased resistivity) observation, geomorphological circumstances such as the relief and increased creeping velocity of the active rock glacier, are discussed by using additional remote sensing data. The present study highlights ice-poor permafrost degradation in the Alps resulting from ever-accelerating global warming.



## 1 Introduction

Permafrost as a thermal phenomenon of perennially cryotic ground (Ballantyne, 2018) responds directly to atmospheric temperature increase by warming, and, subsequently thawing (Biskaborn et al., 2019; Smith et al., 2022). Significant socio-

economic impacts are projected on human livelihoods as a result of mountainous permafrost degradation (Beniston et al., 2018; Hock et al., 2019). Perennial subsurface ice content acts as a significant fresh-water resource although the time-scales of melting and corresponding runoff contributions are still unclear (Hilbich et al., 2022; Jones et al., 2018; Villarroel et al., 2021). Rock and debris-slope instability results in increased rock-fall and debris-flow frequency and intensity, as well as landslide activity (Chiarle et al., 2021; Deline et al., 2015; Frauenfelder et al., 2018; Haeberli et al., 2017; Patton et al.,

2019). These permafrost-related natural hazards threaten infrastructure, habitants, tourists, and alpine sport activities in high alpine regions (Duvillard et al., 2019; Keuschnig et al., 2017; Mourey et al., 2021).

Systematic alpine-permafrost observations have a long tradition in the Alps (e.g. Haeberli et al., 1993). A Swiss network of thermistors installed in boreholes directly monitors ground temperatures giving the mean annual ground temperature (MAGT) (Etzelmüller et al., 2020; Vonder Mühll et al., 2008), which can be used to distinguish ice-poor from ice-rich

permafrost (Haberkorn et al., 2021; Kenner et al., 2019). As the installation of boreholes is cost and time intensive, indirect methods such as geophysical measurements are often applied for subsurface ice detection and permafrost characterisation (Vonder Mühll et al., 2002). The application of direct-current (DC) soundings in early studies (Barsch and King, 1989; Haeberli, 1979; King et al., 1987) and of electrical resistivity tomography (ERT) more recently (Hauck, 2002; Kellerer-Pirklbauer and Kaufmann, 2018; Kneisel and Hauck, 2008) has enabled the localisation and characterisation of permafrost in

a variety of alpine settings. ERT measurements differentiate ice-containing debris and bedrock from unfrozen material based on the significantly increased electrical resistivity of frozen material (Hauck and Kneisel, 2008; Mewes et al., 2017). As electrical current conduction in subsurface materials is mainly promoted by water filling the pore space or covering grain surfaces, the electrical resistivity (i.e., the inverse of the electrical conductivity) strongly increases as the aqueous phase freezes out (Barsch, 1996; Hauck, 2002; Hilbich et al., 2009; Mewes et al., 2017; Mudler et al., 2019).

ERT measurements have also been successfully used to detect permafrost on periglacial landforms covered by coarse debris such as protalus ramparts (e.g. Fukui, 2002; Scapozza, 2015), talus slopes (e.g. Kenner et al., 2017; Kneisel et al., 2000; Lambiel and Pieracci, 2008; Scapozza et al., 2011), and rock glaciers (e.g. Buckel et al., 2022; Frauenfelder et al., 2008; Halla et al., 2021; Kellerer-Pirklbauer and Kaufmann, 2018; Maurer and Hauck, 2007; Villarroel et al., 2021). ERT surveys have also been applied to study bedrock permafrost on several mountains in the alps (Keuschnig et al., 2017; Krautblatter et

al., 2010; Krautblatter and Hauck, 2007; Magnin et al., 2015, 2017). Some of these latter studies employed permanently installed ERT profiles to monitor changes in the ground ice content and the active-layer thickness over several years. ERT applications in high alpine areas (Hauck, 2002; Hilbich et al., 2011; Supper et al., 2014) or polar environments (Farzamian et



al., 2020; Oldenborger and LeBlanc, 2018; Uhlemann et al., 2021) have shown that repeated ERT data are capable of monitoring the long-term permafrost evolution. Monitoring intervals can vary from hourly to multi-year intervals, depending
on the research objective, whereby longer intervals are suitable for recording long-term permafrost changes due to climate change (Etzelmüller et al., 2020; Kneisel et al., 2014). Emerging problems such as changing contact resistances, different instruments or inversion artefacts can be solved, for example, by statistical analysis (Supper et al., 2014) or adapted data processing schemes (Oldenborger and LeBlanc, 2018).

However, quantitative analyses of mountainous permafrost degradation related to subsurface ice content are scarce (e.g.
Hilbich et al., 2008; Mollaret et al., 2019) or do not exist at all, especially for longer time periods (>10 a). We fill this gap by repeating ERT measurements along existing transects from the years 2005 to 2011 published in Frauenfelder et al. (2008) and Otto et al. (2012). Compared to regular and continuous measurements, for example by automated ERT systems (e.g. Farzamian et al., 2020; Hilbich et al., 2011; Supper et al., 2014), this is a cost-effective application that can provide a high gain in knowledge. Applying a direct comparison of historical and re-measured ERT profiles, our study aims to identify
warming induced permafrost degradation on periglacial landforms in two regions of the European Alps over the past decade. Conducting the measurements at the very same location of the historical ERT profiles allows for a quantitative detection of degradation-induced resistivity changes as well as for a comparison of historical and re-measured resistivities.

Since alpine permafrost landforms are often difficult to access due to their remote location in high mountains, several challenges arise: If no infrastructure such as roads, cable cars or airborne transport is available, the heavy weight of the
equipment has to be transported by human power (Frauenfelder et al. 2008; Buckel et al., 2021; Halla et al., 2021; Villarroel et al., 2021). The standard measuring equipment includes the control unit, cables, steel pikes, sponges, salt, hammers, canisters, and an adequate amount of water depending on subsurface moisture and the number of electrodes needed for the measurement. Additionally, many permafrost-related ERT surveys were conducted on coarse and blocky surfaces or directly on bedrock lacking vegetation cover, which increases contact resistance. Establishing an optimal galvanic coupling on these
hard surfaces is challenging as pointed out in former studies (Buckel et al., 2021; Hauck et al., 2003; Maurer and Hauck, 2007; Schrott and Sass, 2008; Vonder Mühll et al., 2002). An insufficient galvanic coupling results in high contact resistances, which reduce or completely prevent the current flow into the subsurface. Establishing an optimum galvanic coupling on dry and blocky surfaces, as it is the case with rock glacier and talus-slope surfaces, is very effortful and time-intensive when sponges soaked in salt water are used by wedging them between the blocks (Hauck and Kneisel, 2008). To
address these practical challenges, the present study tests the application of electrodes manufactured from an electrically conductive textile. The use of these newly-developed textile electrodes aims at (1) improving galvanic coupling by increasing the electrically conductive contact area with the surface, (2) reducing equipment weight, and (3) reducing the effort when placing/hammering conventional electrodes (steel pikes).



## 2 Study areas

Three different sites were selected in collaboration with the team of the newly initiated International Database of Geoelectrical Surveys on Permafrost (IDGSP) (https://www.unifr.ch/geo/cryosphere/en/projects/permafrost-monitoring-and-dynamics/idgsp.html), which aims at promoting and coordinating the repetition of historical ERT surveys on permafrost worldwide (Hauck et al., 2020). Three investigated periglacial landforms are located in two regions of the Alps. In the Corvatsch catchment (Eastern Swiss Alps) we re-measured an ERT transect on a protalus rock glacier and evaluated the

performance of the newly developed textile electrodes. The Gianda Grischa rock glacier (Eastern Swiss Alps) represents a test site of different creeping behaviour and a complex geomorphological background. The rock glacier consists of an active and an inactive lobe, where two profiles on each were measured as repetition measurements. The third study site is located on a periglacial talus slope in a different climatic region in the Austrian Alps. The descriptions of the study areas are reduced to information relevant for the interpretation of the results.

### 2.1 Corvatsch Catchment

The Corvatsch catchment is located on the northern slope of Piz Corvatsch (3451 m a.s.l.), in the Upper Engadin, and includes the Murtèl, Marmagnun, and Chastelets rock glaciers. The catchment, which is surrounded by a steep northwest facing headwall, is situated at about 2700 m a.s.l. and can be regarded as one of the best-investigated permafrost sites in the European Alps, with several long-term ERT observations (e.g. Mollaret et al., 2019; PERMOS, 2021; Schneider et al., 2012)

and borehole measurements (e.g. Haeberli et al., 1993; Vonder Mühll and Haeberli, 1990). Climate data is available from a micrometeorological station at the Murtèl rock glacier. Between 1997 and 2018 the MAAT (Mean Annual Air Temperature) was -1.66 °C, with a long-term temperature increase of about 0.7 °C during this period (Hoelzle et al., 2021). The protalus rock glacier on a talus slope (46° 25' 38" N; 9° 48' 54" E) investigated in this study is located north of the Chastelets rock glacier. A steep front and a relatively small flat body characterize the periglacial landform (Figure 1). The substrate layer

consists of small-grained material at the front and inhomogeneous boulders of differing size in the plateau area. Lithological granodiorite and metamorphosed basalt of the Corvatsch nappe as well as muscovite and calcite marble of the Chastelets series are predominant at the study site (Schneider et al., 2013). In August 2011, first ERT measurements were conducted along a 380 m long profile, which was aligned with the creeping direction of the protalus rock glacier (S. Schneider/M. Hölzle/C. Hauck, unpublished). Yellow markings from 2011 were used to tag the profile for easier relocation. The historical

profile was re-measured at the end of August 2021 during dry and sunny weather conditions. This profile is also used to compare the performance of the different coupling methods under field conditions (i.e., steel pikes versus textile electrodes).





**Figure 1: Location of the ERT profile on the protalus rock glacier in the Murtèl catchment, Upper Engadin. The repeated profile (2021) is based on the blocks marked in the field in 2011; therefore the location of the new and the historical profiles are identical.**

## 2.2 Gianda Grischa rock glacier

The Gianda Grischa rock glacier (46° 29' 24" N; 9° 44' 41" E) in the Upper Engadin (Eastern Swiss Alps) was investigated in 2005 using ERT, among other methods (Frauenfelder, 2005; Frauenfelder et al., 2005, 2008). This double-tongued rock glacier is located on the western slopes of Piz Julier (3380 m a.s.l.). The headwall provides debris consisting of granites, diorites, and para-gneiss as a part of the Err-Bernina nappe (Spicher, 1980). Three parts of different activity stages are described by Frauenfelder et al. (2008): An active part (labelled "A" in Figure 2), an inactive part (not visible in Figure 2) and a possibly relict part (B in Figure 2). The active rock glacier, which is located at an elevation between 2540 and 2800 m a.s.l., has a length of 1000 m and a width of 170 to 390 m. The active part is characterised by a steep front as well as furrows and ridges, which are typical features of active rock glaciers (Barsch, 1996). The inactive part of the rock glacier, south of the active part, is largely overridden by the active rock glacier. Therefore, the length and width (about 100 m) of the inactive part are significantly smaller. Unlike the active part, a considerable amount of vegetation is present on the inactive





part. The surface is mostly fine-grained with some larger boulders in between. Of the five profile lines used for the ERT measurements conducted in August 2005 by Frauenfelder et al. (2008), four tomograms were repeated at the end of August 2021 during dry and sunny weather conditions in this study (Figure 2). The active part hosts the intersecting profiles P2 and P3, the inactive part the intersecting profiles P1 and P4 of the rock glacier.


**Figure 2: Location of the historical (2005, dashed line) and the repeated (2021, solid line) ERT profiles on the Gianda Grischa rock glacier, Upper Engadin. The status of activity (A = active; B = inactive) is based on this study and Frauenfelder et al. (2008). The area around A is characterized by furrows and ridges and hosts profiles P2 and P3.**

## 2.3 Glatzbach catchment

The Glatzbach catchment in the Tauern range (central Austrian Alps) was surveyed by Otto et al. (2012) in 2008 and 2009. These authors used a multi-method approach including ERT measurements to detect mountain permafrost. The investigated northeast-exposed talus slope (47° 2′ 22″ N; 12° 42′ 33″ E) ranges between 2700 and 2900 m a.s.l. in elevation with an average inclination from 25° to 45°. The talus slope is characterized by a coarse-grained block field consisting of quartzite



and dolomitic marble. The surrounding areas are dominated by a fine-grained, phyllitic substrate layer originating from the weak "Matreier Schuppenzone". This zone forms the transition from the "Obere Schieferhülle" (Penninic) in the north to the crystalline basement (Austroalpine) in the south (Stingl et al., 2010). The MAAT at an elevation of 2650 m a.s.l. increased between 1997 and 2010 from -2.5 °C to -1.2 °C by 1.3 °C (Otto et al., 2012).

Two ERT profiles of 96 m length each, measured in July 2009 by Otto et al (2012), were repeated in September 2021 during
sunny weather but wet surface conditions due to melting snow. GL-P1 cuts in E-W direction through the block field, while the intersecting profile GL-P2 runs in N-S direction (Figure 3).

**Figure 3: Location of the historical (2009, dashed lines) and the repeated (2021, solid lines) ERT profiles on the block field in the**
**Glatzbach catchment, Tauern range.**



# 3 Methods and Data

## 3.1 Electrical resistivity tomography (ERT)

The electrical resistivity of the subsurface is measured using a quadrupole setup: Two electrodes are used to inject an
electrical current into the ground. This current generates an electrical field, which depends on the spatial variation of
electrical resistivity in the subsurface. The characteristics of this electrical field are then captured by measuring a potential
difference (voltage) between another pair of electrodes. From the measured potential difference, the injected current, and a
geometric factor, which relates the measured voltage to the quadrupole geometry, an apparent resistivity of the ground below
the quadrupole can be calculated, which represents an average resistivity of the half space below the quadrupole. The
investigation depth scales with the total length of a linear array of electrodes while the (near-surface) resolution is higher for
short electrode spacings. The Wenner- configuration used for the historical and the repeated measurements provides a good
signal-to-noise ratio combined with a good resolution of horizontal structures (Kneisel and Hauck, 2008). In this
configuration, the distances between the four electrodes of the individual quadrupole measurements are all the same. In order
to obtain information on the 2D variation of electrical resistivity below the profile, the position of the four-electrode setup
along the line has to be varied to assess the lateral variation of resistivity and the array length has to be varied to resolve
vertical variations (Kneisel and Hauck, 2008).

The processing of such a 2D data set includes two steps: filtering and inversion. In the first step, the data quality of each
quadrupole measurement is checked and those measurements with insufficient data quality are removed from the data set.
We used two criteria for filtering: (1) all data points with a relative standard deviation of all stacks (repeated measurements)
above 0.3 % and (2) all data points with the maximum value of measurable voltage possible by the device, which indicates
that no sufficient signal strength can be achieved, were removed. Additionally, a defect in one cable produced random values
for measurements with one specific electrode. These values were deleted and reconstructed by interpolating the neighboring
values. Visually identifiable outliers were also deleted. The historical data were only filtered manually using the
"Exterminate bad datum points"-function in the RES2DINV software. Bad data points are values with apparent resistivity
that are apparently too large or too small compared to the neighbouring values. The overall number of filtered data points is
given in tables 2-4 for the three surveyed catchments.

The measured apparent resistivity corresponds to the true resistivity only for a homogenous substrate. Therefore, in the
second step, so-called inversion methods have to be applied to model the spatial variation of an inhomogeneous substrate in
the subsurface (Kneisel and Hauck, 2008). We used the RES2DINV software package, which is based on a smoothness-
constrained least-squares inversion process (Loke and Barker, 1995) to obtain ERT tomograms from the measured data.
During this iterative process, the differences between the measured apparent resistivities and the modelled ones are
minimized by adjusting the resistivities assigned to discrete blocks representing the subsurface. The remaining discrepancy
between measured and modelled apparent resistivity values is expressed in terms of the Root Mean Square (RMS) error.



Besides the ice content, other characteristics, such as lithology, porosity, water content, grain size, packing density, etc., control the resistivity of the subsurface. Consequently, the resistivity of permafrost varies over a wide range (Hauck and Kneisel, 2008; Herring and Lewkowicz, 2022). Here, we describe the subsurface for a more detailed interpretation by the use of common terms like unfrozen, ice-poor permafrost and ice-rich permafrost. The resistivity anomalies regarding frozen and unfrozen sediment are visually distinguishable in the ERT tomograms. A clear definition of a threshold resistivity for the transition between unfrozen and frozen conditions is difficult due to the factors mentioned above, but many previous studies assume a rough value of 10 kΩm as orientation to distinguish frozen from unfrozen sediment (bedrock permafrost excluded: Halla et al., 2021; Hauck and Kneisel, 2008; Mewes et al., 2017; Otto et al., 2012; Palacky, 1988). It is however strongly subjective and should be adjusted towards much lower values in the case of conductive lithology or towards much higher values in the case of, e.g. coarse-blocky and very porous substrate. Ice-poor permafrost may occur in terms of patches or larger zones of perennial ground ice. Ice-rich permafrost is characterised by high resistivity values (usually >150 kΩm) in a massive ice body. The interpretation of the ERT tomograms considers the different regolithic subsurface properties for the upper Engadin (Gianda Grischa and Corvatch) and the southern edge of the Tauern Window (Glatzbach catchment). The colour scale of resistivity was determined for each homogeneous subsurface individually. Similar ice content and similar lithology of the subsurface determine the homogeneity. The non-logarithmic classification is based on a higher resolution in the low resistivity range, since larger variations and anomalies are expected within this range. However, to ensure direct comparability, the historical and the recent tomograms are displayed using identical colour scales and classifications.

**3.2 Location and repetition of historical ERT measurements**

Since the main goal of this study is to compare the recent and historical measurements, the historical profile locations had to be correctly located in the field. The IDGSP (Hauck et al., 2020) provides GPS data and basic information on coordinate system and precision. In preparation of the 2021 field work, historical GPS data were plotted on a map, which contained additional information based on high-resolution orthophotos (Bing maps) and terrain analyses from a digital elevation model (Otto and Smith, 2013). The orthophotos visualize large boulders, which helped to locate the start and end points in the field. Highest priority was given to localizing the centre point of the historical profile, as (due to technical constraints) we could not always use the same geometrical setup in terms of total number of electrodes and electrode spacing as in the historical measurements. Like the data collectors of the historical ERT profiles, we used a non-differential GPS. Therefore, the accuracy of the historical as well as the newly measured location data is within the range of a standard handheld GPS (Garmin eTrex 10 used in 2021), i.e., depending on the satellite availability roughly 5-10 m in horizontal direction.

The availability of historical markers facilitated a proper localization in the field: Stone cairns, historical GPS data and yellow strokes on prominent blocks were available as historical markers at the Swiss study sites. Historical profiles in the Austrian Glatzbach catchment were only documented by GPS locations. In addition, easily identifiable markers like huge boulders in aerial photographs were also used to localize the profile lines in the field. At the Gianda Grischa rock glacier,



cairns were still present at the start and end point as well as the centre point of the profiles. In the Corvatsch catchment, yellow markers indicated the location and orientation of the historical 380-m long profile. An exact localization of the start, centre and end point along the profile was, however, not possible due to the absence of historical GPS coordinates. Instead, prominent topographic features were used to relocate the profile along the along the profile direction. The length of the

historical tomogram was shortened by 132 m at the beginning and 52 m from the end to yield a 196 m long profile for enhanced comparability.   In the Glatzbach catchment, due to the absence of any markers in the field, only the relatively rough historical GPS locations were available to relocate the historical profile. Generally, profile topography was estimated by measuring the differences between consecutive electrode positions with a yardstick.

The Glatzbach catchment contains two historical ERT profiles, which were measured in July 2008. These were repeated

during early September 2021 (Table 3). The fact that the profiles were measured at different times challenges the interpretation due to the large seasonal variability close to the surface. Therefore, we focus the discussion of the results on depths below the assumed active layer to exclude misinterpretations of seasonally conditioned resistivity changes. The value of 10 kΩm to distinct the active layer from the frozen part is given by Otto et al. (2012).

### 3.3 Measurement setup and newly developed textile electrodes

For this study, we used the "GeoTom-MK" (GEOLOG2000, Augsburg, Germany) with a unit electrode spacing of 4 meters and a total of 50 electrodes resulting in a maximum profile length of 196 m. The instrument automatically determines the optimum input current by predefined levels (0.005, 0.05, 0.5, 5.0, 50.0 mA). A higher required current indicates (1) poorer coupling conditions due to higher contact resistance or (2) high subsurface resistivity. Conventionally, a combination of stainless steel rods and watered or salt-water soaked sponges, which are placed between individual blocks, are used on

debris-covered surfaces (e.g. Hauck et al., 2003). On rock faces or large bedrock surfaces, electrodes can be drilled into the rock, e.g. by using climbing bolts and copper contact paste (Krautblatter and Hauck, 2007). Although yielding sufficient galvanic contact, this approach is effortful and involves an invasive intervention by drilling into the ground material. However, in very coarse substrate with huge boulders, both approaches can be very time-consuming and good galvanic coupling cannot always be achieved.

In the present study, we test the performance of a newly developed alternative electrode made of conductive textile. The commercially available textile (Holland Shielding Systems BV) consists of polyester with woven-in copper and nickel threads. The cost of one textile electrode made of this material is ~15 €, manufacturing one electrode takes about 10 min. We cut the textile into square pieces (30 cm x 30 cm) and form small bags, which are filled with 200 grams of fine sand (e.g. available in the field as glacial till) and closed by a cable tie (Figure 4). The fine sand is wrapped in plastic film, which

prevents moistening of the sand. The sand weighs down the bag, so that the conductive textile is pressed onto the ground and the contact area is maximized. The key parameter for the electrical coupling between electrodes and subsurface material is





the contact resistance (e.g. Hördt et al., 2013; Rücker and Günther, 2011), which is controlled by the electric contact area of the two materials. To ensure sufficient current flow and thus sufficient signal quality, high contact resistances can be further reduced by placing wet sediment and sediment containing biotic material (e.g. roots, moss) between the textile electrode and

the rugged surface of boulders. Additional salted water further reduces contact resistances of the textile electrodes and improves required contacts between the boulders in the near subsurface (e.g., the active layer). However, a generally poor electric contact between large blocks in the subsurface cannot be compensated by the use of the textile electrodes.

We chose the protalus rock glacier in the Corvatsch catchment to test and evaluate the textile electrodes due to its smooth profile topography as well as the easy retrieval of the historical profile location by yellow markers. The measurement setup

consists of a first baseline measurement using steel pikes in moist ground. In the second measurement we placed wet sponges at the same location and the third measurement was conducted with textile electrodes. The data were analyzed to assess the performance of the textile electrodes for subsequent measurements. The (successful) evaluation of the textile electrodes at this first test site resulted in a mixed application of textile and steel electrodes at the subsequent profiles on the Gianda Grischa rock glacier and in the Glatzbach catchment, depending on the surface characteristics. Blocky surfaces were

investigated using textile electrodes, fine-grained and vegetated surfaces by steel pikes.



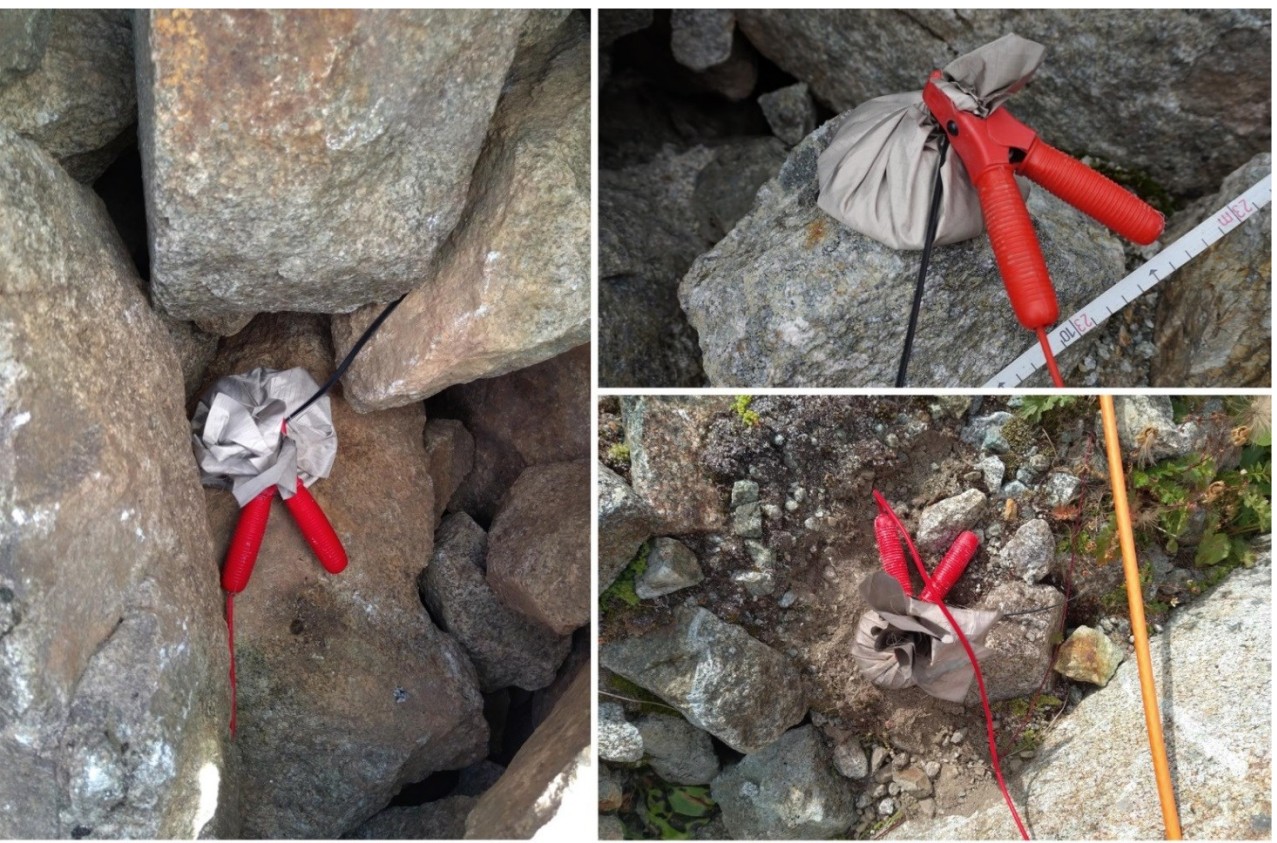

**Figure 4: Prototype of the textile electrodes in use. The bag made of conductive textile is filled with 200 g of fine sand. The electrode is connected to the measuring cable by the red clip.**

## 4 Results and Interpretation

### 4.1 Textile electrodes

In order to investigate the data quality obtained with the new textile electrodes, we directly compare ERT measurements (Figure 5) carried out with steel pike electrodes connected to the ground by watered sponges (Figure 6, (c)) and the textile electrode (Figure 6, (d)). For the quantitative analysis, the apparent resistivity, the relative standard deviation, and the input voltage of the device were compared for all 377 quadrupoles of the measurement. 15 quadrupoles result in unrealistic values

due to a cable defect, which are not considered in the quantitative comparison.

In a first step, all quadrupole measurements were filtered according to the criteria defined in chapter 3.1. On average, the (unfiltered) data collected with conventional electrodes show slightly higher values of the required input current for conventional (1.0 mA) than for the textile electrodes (0.7 mA) as well as for the relative standard deviation (7.8 % vs. 6.7 %). Consequently, more data points collected with steel pike electrodes had to be removed (6 %) than for the textile

electrodes (4 %). The percentage of removed data itself is comparable to multi-channel measurements by earlier studies on



harsh surface conditions of a rock glacier (e.g. Halla et al., 2021). In summary, these first observations (i.e., differences in input current, standard deviation, and percentage of removed data) indicate a slightly higher quality of the data collected with the new textile electrodes.

From a more practical point of view, both electrode types – if placed at the very same locations – should at least result in the
same measured apparent resistivity values. Therefore, we also analyze the relative difference between the two measured apparent resistivities. The median of the relative difference in the apparent resistivity is 2.1 %. Furthermore, the histogram in Figure 5 shows that most of the deviations are small and only few outliers with larger discrepancies exist. Only 10 % of all compared quadrupoles show relative differences >14 %.

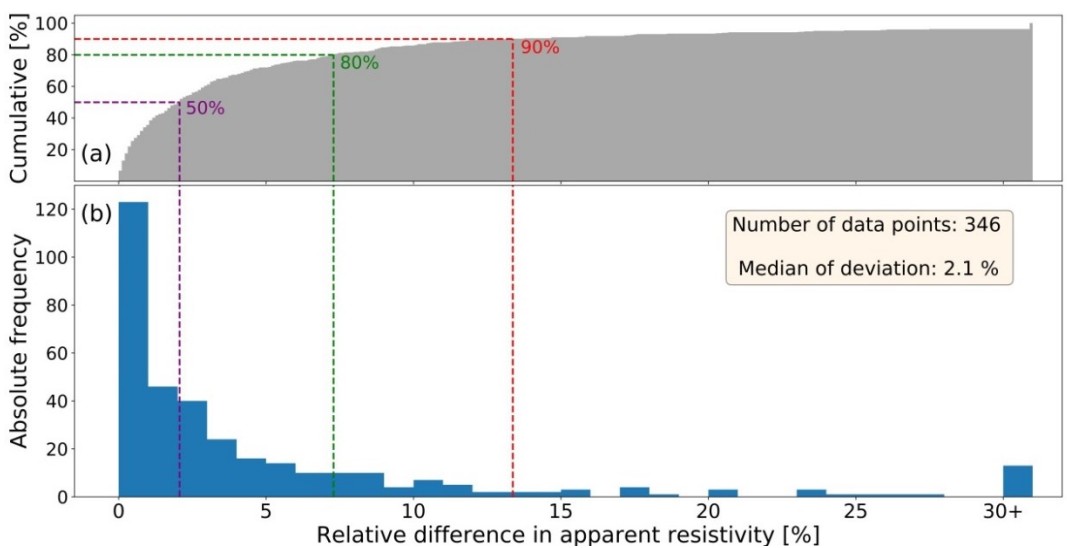

**Figure 5: Histogram of the percentage deviation of the measured apparent resistivity for pairwise comparison between quadrupole measurements collected with the new textile electrodes and conventional steel pike electrodes: (a) cumulative distribution and (b) absolute frequency (b). The dashed lines show defined portions of all data pairs with the corresponding limit of deviation in apparent resistivity.**

Nevertheless, the comparison of both electrode types indicates sufficiently good comparability of the measurements with
reliable and even slightly better results for the textile electrodes, hence justifying the application of the novel textile electrodes at the other survey sites.

### 4.2 ERT-based permafrost detection

### 4.2.1 Corvatsch catchment

Four ERT profiles are presented from the protalus rock glacier at the Corvatsch catchment (Table 1; Figure 6). The original
survey from August 18, 2011 was repeated on August 25, 2021 in order to assess the temporal evolution of the permafrost in this profile over 10 years and to further compare the data collected with different electrode types. The inversion yields a RMS error of 4.6 % for the historical data (Table 2). The RMS error of the repeated data varies between 5.5 and 5.9 %





despite the different electrode types used. Additionally, the visual comparison of the three tomograms representing different
electrode types only shows small deviations and an overall reliable fit of the resistivity distributions (Figure 6, (b) to (d)).
This observation further confirms the applicability of the newly developed textile electrodes, which nevertheless required
additional supply of water to reduce contact resistance between larger blocks at the surface.

**Table 1: Overview of the historical and repeated ERT profiles of the Corvatsch catchment. In 2021, the same profile was measured three times using different electrode types.**

| Date of data acquisition | Electrode type | No. of electrodes | Electrode spacing [m] | Instrument | No. of iterations | RMS error [%] | No. of filtered (unfiltered) data points |
|---|---|---|---|---|---|---|---|
| **18.08.2011** | Steel pikes | 50 | 4 | SYSCAL | 5 | 4.6 | 345 (345) |
| **25.08.2021** | Steel pikes | 50 | 4 | GeoTom-MK | 4 | 5.9 | 356 (377) |
| **25.08.2021** | Steel pikes, wet sponges | 50 | 4 | GeoTom-MK | 4 | 5.5 | 354 (377) |
| **25.08.2021** | textile | 50 | 4 | GeoTom-MK | 4 | 5.9 | 361 (377) |

The historical tomogram display highest resistivity values of around 40 kΩm (bluish colour between 32-56 m) (Figure 6, (a))
which indicate presumably ice-poor permafrost zones that are less pronounced in the repeated profile(s). Here, the maximum
resistivity has reduced to 30 kΩm (green colours in Figure 6 (b) to (d)), suggesting an overall decrease in the ice content of
the protalus rock glacier. We also note an overall decrease of the resistivity within the active layer between 2011 and 2021,
and a general thickening of the active layer around the centre of the tomogram.








**Figure 6: Tomograms of the historical (2011) and the repeated (2021) profiles in the Corvatsch catchment: (a) historical profile; (b) repeated profile measured with steel pikes; (c) repeated profile measured with steel pikes and wet sponges; (d) repeated profile measured with textile electrodes. The black lines in (b) to (d) indicate the outline of the historical tomogram. The vertical axes display the relative elevation in meters.**

### 4.2.2 Gianda Grischa

Four historical ERT profiles on the Gianda Grischa rock glacier collected on August, 29 2005 were repeated on August, 26-28, 2021 (Table 2) using a combination of watered steel pikes and textile electrodes. Differences in the number of electrodes and electrode spacings between the historical and repeated measurements (Table 2) result in different lengths of the historical and repeated profiles. The RMS error of the inversion results is significantly smaller for all newly measured data compared to the historical data, for a similar number of iterations. This difference in data quality is probably due to significantly different surface and weather conditions (measurements in 2005 were obtained under very dry surface conditions). We suspect that the larger contact interfaces of the textile electrodes compared to the steel pikes also added to the better data quality.





**Table 2: Overview of the historical and repeated ERT profiles on the Gianda Grischa rock glacier. The second column indicates**
**historical ("H") and newly measured data ("N").**

|    |   | Date of data acquisition | Number of electrodes | Electrode spacing [m] | Instrument | Number of iterations | RMS error [%] | No. of filtered (unfiltered) data points |
|----|---|---|---|---|---|---|---|---|
| **P1** | H | 29.08.2005 | 36 | 5 | Syscal | 4 | 10.8 | 167 (198) |
|    | N | 28.08.2021 | 50 | 4 | GeoTom-MK | 4 | 7.1 | 345 (392) |
| **P2** | H | 29.08.2005 | 36 | 5 | Syscal | 4 | 14.7 | 158 (198) |
|    | N | 27.08.2021 | 50 | 4 | GeoTom-MK | 3 | 10.6 | 392 (392) |
| **P3** | H | 29.08.2005 | 36 | 5 | Syscal | 4 | 16.3 | 173 (198) |
|    | N | 26.08.2021 | 50 | 4 | GeoTom-MK | 4 | 7.3 | 360 (392) |
| **P4** | H | 29.08.2005 | 36 | 5 | Syscal | 4 | 8.3 | 182 (198) |
|    | N | 28.08.2021 | 50 | 4 | GeoTom-MK | 4 | 7.9 | 383 (392) |

Profiles GG-P1 and GG-P4 are mainly located on the inactive part of the rock glacier (Figure 2), which is characterized by significantly lower overall resistivity values (< 50 kΩm, cf. reddish colour in Figure 7 and Figure 8) compared to the higher values observed in the active part of the rock glacier (left-hand parts of GG-P1 and GG-P4, as well as profiles GG-P2 and GG-P3 (> 100 kΩm, blue colour in Figure 9 and Figure 10). The latter correspond to the central lobe of the active part of the
rock glaciers. This general pattern in the resistivity distribution did not change with time, although a general resistivity decrease is observed.



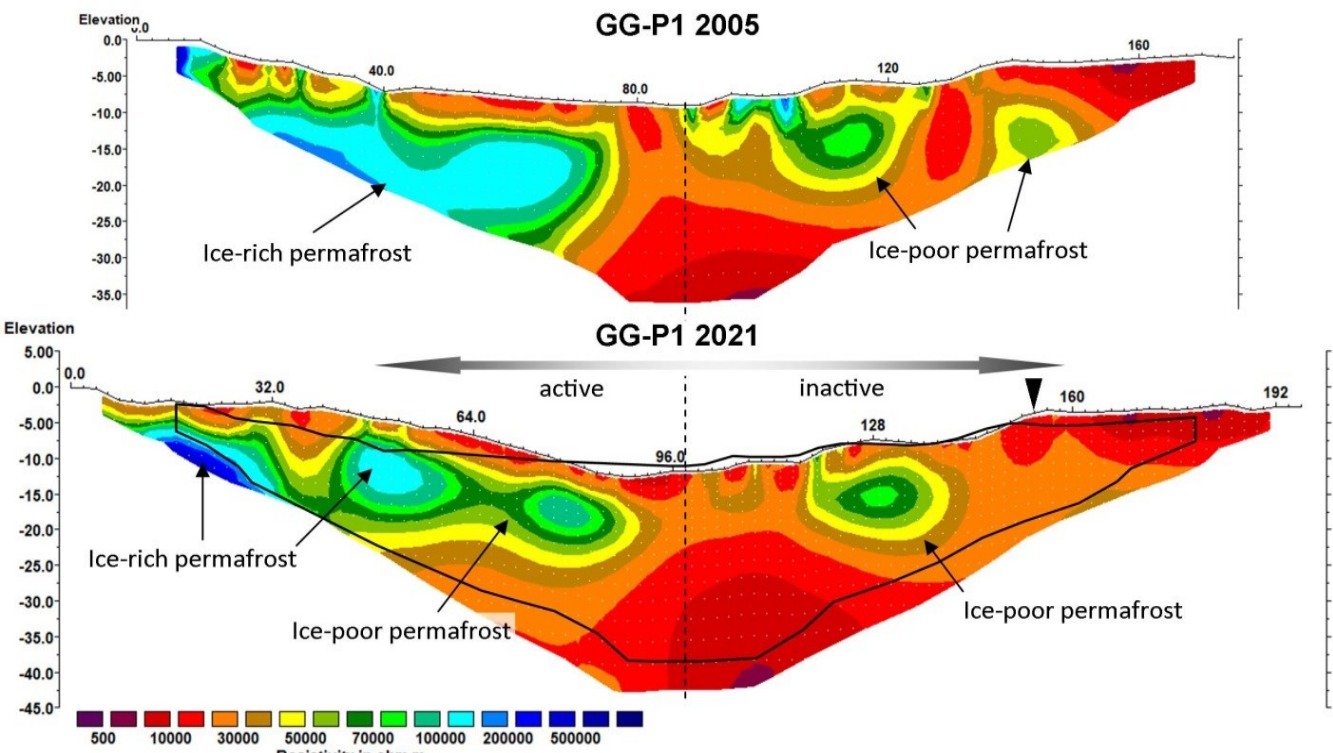

**Figure 7: Tomograms of the historical (2005) and the repeated (2021) profile GG-P1 on the Gianda Grischa rock glacier. The dashed line indicates the common centre point. The point of intersection with tomogram GG-P4 is indicated by the black triangle.**
**The black line indicates the outline of the historical tomogram.**

Profile GG-P1 starts at a central lobe on the active part of the rock glacier and continues towards the inactive part to the South (Figure 2). Both profiles, the historical and the repeated, show resistivity values higher than 200 kΩm for the part of the profile located on the active rock glacier (left part of Figure 7). Below the first 80 m of the historical profile, resistivity values lower than 50 kΩm at shallow depths <5 m represent the active layer. This surface layer is heterogeneous with more

resistive zones (representing dryer conditions) between 0 and 15 m and around 40 m. The overall resistivity of this surface layer decreased between 2005 and 2021. At larger depths, the historical profile shows resistivity values around 150 kΩm, which we interpret as ice-rich permafrost with a thickness of ~15 m. In the 2021 tomogram, this formerly compact resistive anomaly became patchier with an overall decrease in resistivity values which might indicate a transition from massive ice-rich permafrost to ice-poor permafrost.

Similarly, an overall resistivity decrease between 2005 and 2021 is also observed in the second half of the tomogram between 2005 and 2021. Two more resistive anomalies with high resistivity values (>50 kΩm) appear around 110 m and 140 m along the profile. In 2021, the resistive anomaly around 140 m has completely disappeared, while the other anomaly displays reduced resistivity values. We conclude that ice-poor permafrost degraded strongly during the investigation period.





**Figure 8: Tomograms of the historical (2005) and the repeated (2021) profile GG-P4 on the Gianda Grischa rock glacier. The dashed line indicates the common centre point. The point of intersection with tomogram GG-P1 is indicated by a black triangle. The black line displays the outline of the historical tomogram.**

Profile GG-P4 stretches in W-E direction, starting on the active part and extending into the inactive part of the rock glacier (Figure 2). It is intersected by P1 at 124 m. The highest resistivity values (>150 kΩm) are observed on the still active part (Figure 8), suggesting ice-rich permafrost. A highly resistive anomaly of values with up to 100 kΩm, which is clearly visible in the historical tomogram around 80 m along the profile, has strongly reduced in size in the 2021 tomogram, indicating melting of formerly ice-poor permafrost. However, the proximity to the surface also allows the interpretation that the resistivity anomalies relate to air voids due to ice melt. Below ~10 m depth, values <10 kΩm characterize the remaining part of the repeated profile as unfrozen. In the historical tomogram, the frozen area partly extended down to >15 m depth. This clearly indicates a degradation of permafrost during the 16-yer period.

The main lobe of the rock glacier hosts profiles GG-P2 (Figure 9) and GG-P3 (Figure 10). These differ significantly from profiles GG-P1 and GG-P4 by a roughly two-layered structure and resistivity values of >1 MΩm. The upper layer represents




the coarse-blocky active layer with resistivity values of <50 kΩm. Below, the extremely high resistivity values of >200 kΩm indicate the presence of massive ice.



**Figure 9: Tomograms of the historical (2005) and the repeated (2021) profile GG-P2 on the Gianda Grischa rock glacier. The thin black dashed line indicates the common centre point. The point of intersection with tomogram GG-P3 is given by a black triangle. The thick dashed line and the arrow highlight the observed down-gradient shift of furrow. The black line displays the outline of the historical tomogram.**

Lower resistivity (below ~50 kΩm) characterizes the unfrozen parts as the active layer (dark red colour in Figure 9). This active layer consists of coarse and blocky boulders with air-filled interstices. The 2021 measurement shows a smaller overall resistivity of the active layer as well as indications for a general thickening. At a depth of ~25 m resistivity strongly decreases to < 100 kΩm indicating the lower boundary of the ice-rich permafrost, and a thickness of the massive ice core of ~15 – 20 m. Interestingly, from 2005 to 2021 the position of the furrow shifted downward by ~10 m (Figure 9, black arrow). The highly resistive anomaly in the first part of the tomogram shifted downward by ~20 m. In addition, a considerable resistivity increase to values >1 MΩm is observed in the upper part of the profile in the 2021 tomogram between 128 and 164 m. This phenomenon is further discussed in section 5.2.





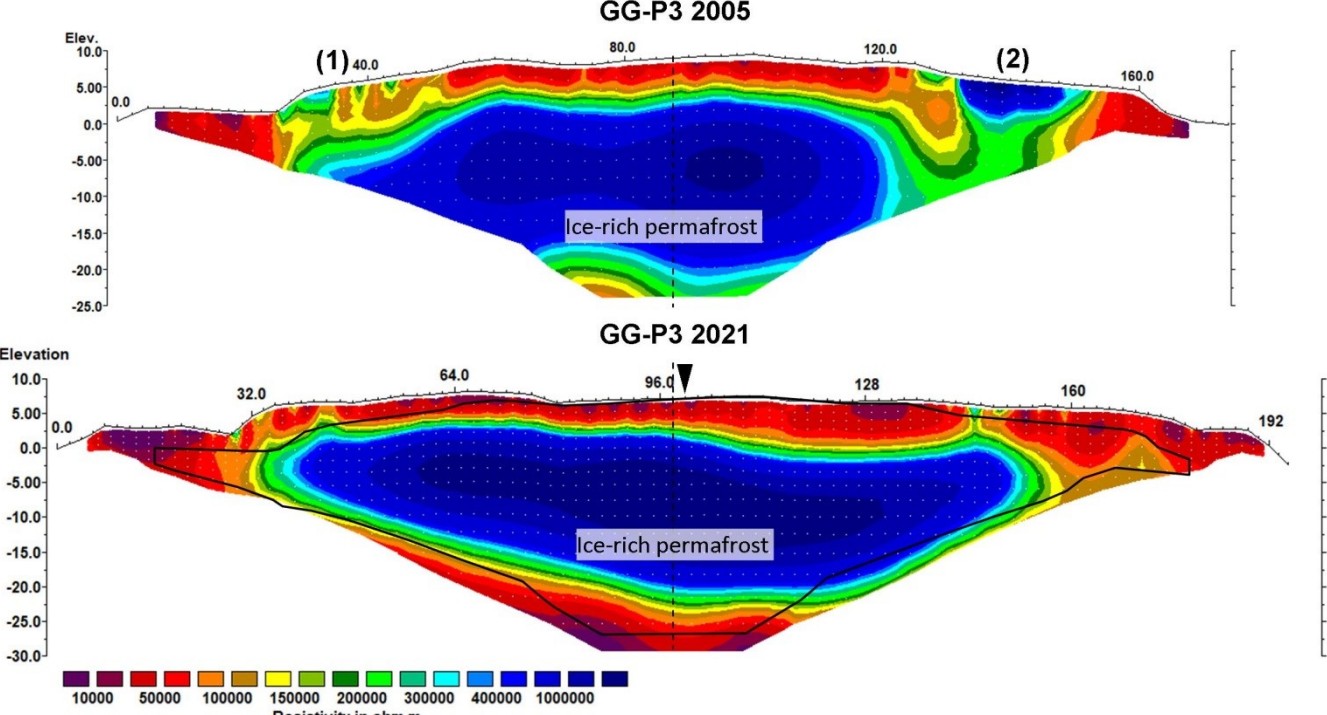

**Figure 10: Tomograms of the historical (2005) and the repeated (2021) profile GG-P3 on the Gianda Grischa rock glacier. The**
**dashed line indicates the common centre point. The point of intersection with tomogram GG-P2 is indicated by a black triangle.**
**The black line displays the outline of the historical tomogram.**

Profile GG-P3 intersects profile GG-P2 at 2615 m a.s.l. parallel to the contours (see Figure 2). The tomogram represents a
cross-section of the active lobe. Similar to the profile GG-P2, the active layer is clearly visible and shows higher and more
heterogeneous resistivity values in the historical data set compared to the repeated profile. This may indicate that
measurement conditions in 2005 were complicated by dryer conditions, which probably caused much higher contact
resistances and an overall poorer data quality. In general, a thickening of the active layer is visible in 2021 compared to
2005.

The very high resistivity values (>1 MΩm) in the central part of both the historical and the new tomograms clearly indicate
ice-rich permafrost or even massive ice. Additionally, the resistivity values in the central section of ice-rich permafrost
increased from 1 MΩm in 2005 to 2 MΩm in 2021, which is also observed in the upper part from profile GG-P2 (cf. section
5.2 for further discussion).

**4.2.3 Glatzbach catchment**

Given the seasonal difference in data acquisition, we limit our data description and interpretation to the area below the
seasonally conditioned active layer. Values around 10 kΩm given by the dark green colour mark the transition from the
unfrozen to the frozen zone in the repeated tomograms (Figure 11).





**Table 3: Overview of the historical and repeated ERT profiles in the Glatzbach catchment.**

|  | Date of data acquisition | Number of electrodes | Electrode spacing [m] | Instrument | Number of iterations | RMS error [%] | No. of filtered (unfiltered) data points |
|---|---|---|---|---|---|---|---|
| **GL-P1** | 04.07.2009 | 25 | 4 | GeoTom-MK | 5 | 3.2 | 92 (92) |
|  | 01.09.2021 | 25 | 4 | GeoTom-MK | 5 | 5.3 | 87 (92) |
| **GL-P2** | 04.07.2009 | 25 | 4 | GeoTom-MK | 4 | 3.8 | 92 (92) |
|  | 01.09.2021 | 25 | 4 | GeoTom-MK | 4 | 8.7 | 84 (92) |

Profile GL-P1 crosses the talus slope in East-West direction. The general pattern of subsurface resistivity is characterized by a section of higher resistivity values at ~5 m depth below the centre, embedded in a more conductive surrounding (Figure 11, 
a). The resistivity values (>30 kΩm) in the centre of the historical tomogram indicate ice-poor permafrost. While the historical tomogram reaches maximum resistivity values of around 80 kΩm, the newly measured profile only reaches maximum values of ~30 kΩm, i.e., the resistivity of the central anomaly of ice-poor permafrost is reduced by around 50 kΩm.

This significant reduction of overall resistivity is also evident in the upslope part of profile GL-P2 (Figure 11, b). While the 
historical profile GL-P2 contains a highly resistive zone with values higher than 30 kΩm (64–82 m), values between 10 and 20 kΩm appear at the same location in the newly measured profile. Below the first half of the profile (0–54 m), values of up to 3 kΩm in the historical tomograms GL-P2 indicate non-frozen substrate. The similarity of the resistivity values below the first half of the profile (0–60 m) at two different times of the year (July 2009 and September 2021) indicates practically constant electrical resistivities of the ice-free subsurface, which highlights the usefulness of ERT tomograms for the 
identification of permafrost degradation over longer periods of time.





**Figure 11: Tomograms of the historical (July 2009) and the repeated (September 2021) profiles GL-P1 and GL-P2 in the Glatzbach catchment. The dashed lines indicate the common centre points. The points of intersection of the two perpendicular profiles are indicated by black triangles. The black lines display the outlines of the historical tomograms.**



## 5 Discussion

### 5.1 Textile electrodes

In order to overcome the difficulties of galvanically coupling electrodes to the ground for ERT measurements in alpine settings, we designed, manufactured and tested textile electrodes under field conditions. Our results show a good (and even slightly better) quality of the apparent resistivity data measured with the textile electrodes compared to conventional steel pike electrodes (chapter 4.1). This is most likely due to the good galvanic contact, which can be established between the flexible material of the electrodes and the uneven surface of outcropping bedrock or loose blocky material covering the surface. It is also possible to combine steel pikes and textile electrodes in a multi-electrode field setup (as realized on the Gianda Grischa rock glacier and in the Glatzbach catchment): simple steel pikes are better suited on fine-grained sediment and/or vegetation-covered surfaces, while textile electrodes are much easier to install and perform better on hard and blocky surfaces. Uhlemann and Kuras (2014) argue that the point-pole approximation usually used in the inversion of ERT data is valid, if the electrode spacing is at least 4 times the diameter of the contact area. Since the diameter of the contact area of our new textile electrode is <15 cm and our minimum electrode spacing is 4 m, we are well within these limits.

In non-geoscientific applications, non-invasive electrodes are widely used for moisture measurements on hard surfaces or for medical applications (e.g., for the recording of electrocardiograms). In geophysics, various materials and geometrical shapes of non-invasive electrodes have already been investigated in other studies, such as small medical electrodes (Sass and Viles, 2006) and metallic mesh electrodes (Tomaškovičová et al., 2016). For capacitive resistivity measurements, where a galvanic contact between electrode and ground is not even wanted, non-invasive electrodes can be considered part of the measurement principle (Kuras et al., 2006; Mudler et al., 2019).

Despite of these similarities, our new textile electrodes differ from these existing non-invasive electrode types. One important aspect is the flexibility of the electrode shape, which optimizes the coupling conditions by maximizing the contact surface between electrode and surface. The electrodes used in this study are prototypes and can still be improved for future studies. Due to the simple processability, different shapes and sizes of electrodes can be realized, which makes this electrode type very versatile and easily adaptable to measurements on different scales (from cm to km). Also, regarding the filling material, the textile electrode is highly flexible.

Besides applications in alpine environments, our new non-invasive textile electrodes will be very useful for electrical measurements on other hard surfaces. In particular, we foresee many interesting applications in urban geophysics, where – due to the type of investigated objects (roads, buildings or infrastructure) – only non-invasive methods are feasible.

### 5.2 Impact of internal deformation on the comparison of new and historical ERT data

In addition to the technical aspects for a proper installation of the historical profile (cf. chapter 3.2), geomorphological conditions must also be taken into account, with respect to the interpretation of permafrost degradation on the basis of the resistivity distribution. The nature of active rock glaciers involves downslope creeping of the debris-ice mixture (Barsch,



1996). Creeping patterns and velocities of rock glaciers vary as a function of their ice content and the slope angle (Haeberli et al., 2006), and are preconditioned by morphometric, topoclimatic and lithological catchment characteristics (e.g. Blöthe et al., 2020; Buckel et al., 2022).

The results from the profiles on the active part of the Gianda Grischa rock glacier (GG-P2 and GG-P3) both reveal higher resistivity values in the massive ice core (Figure 9) in the 2021 tomograms as compared to those measured in 2005. Due to the global temperature increase, which accelerates permafrost degradation and leads to generally decreasing resistivities in the Swiss Alps (Mollaret et al., 2019), we would rather expect a general decrease in resistivity due to the melting of subsurface ice – along most other profiles in this study. We attribute this unexpected increase in resistivity to an effect of a

compressive flow pattern of the ice-rich rock glacier material, which was formerly located further up. Today, this rock glacier material accumulates and compresses in the flatter area of the location of profiles GG-P2 and GG-P3, which reveals itself by the shorter distances between the contour lines on the map in Figure 2.

Analyzing aerial imagery, Frauenfelder et al. (2005) identified an average creep rate of 0.4 to 0.5 ma$^{-1}$ (1971–1998) for the active part of the Gianda Grischa rock glacier (Figure 12, a) with highest creep rates of approximately 1 ma$^{-1}$ (1971–1998)

within these steeper areas (Figure 12, a). The location of the general fastest creep remains in the steeper area of the rock glacier in the most recent time period (2015-2019) while the area of the fastest creep extended downslope (Figure 12 (b)). Additionally, the map illustrates a general increase of velocity in the entire lower part of the rock glacier by a factor of 1.5 to 2. The general pattern of the superficial appearance, such as the rock glacier-typical furrow-ridge sequence, is preserved in the newly obtained topography of the longitudinal profile GG-P2 (Figure 9). In 2021, these prominent topographic features

were shifted ~10 m downwards along profile P2 (Figure 9). This shift fits roughly the creep rate derived by Frauenfelder (2005). The same creeping behaviour can also be observed at the transversal profile GG-P3, which is situated parallel to the contour lines and crosses Profile GG-P2 (Figure 2). Between 2005 and 2021, the central point of profile GG-P2 was shifted down gradient by ~9 m. Especially, the area encompassed by the dotted polygon indicates an acceleration of the average annual creep rate from 0.17-0.26 ma$^{-1}$ (green) to 0.47-0.56 ma$^{-1}$ (orange) due to climate-induced temperature increase (Figure

12 (b), Marcer et al., 2021). The resulting faster creep behaviour supplies the zone marked by the dotted polygon close to profiles GG-P2 and GG-P3 with rock glacier material including ice-rich permafrost, which is reflected by the high resistivity values along the transversal profile GG-P3. This creep behaviour also affected the marked starting points of profiles GG-P1 and GG-P4, and the downslope movement of the markers (cairns) is consistent with the flow pattern (cf. Figure 2).







**Figure 12:** The average annual creep rate in the time period a) 1978-1998 (modified from Frauenfelder et al., 2008) and b) 2015-2019, as obtained from repeated aerial imagery. The arrows encompassed by the dotted polygon illustrate the velocity increase, highlighted by the colour change from green (0.17-0.26 m/a) to orange (0.47-0.56 m/a) between the two observation periods. The black thin lines in (b) indicate the recently measured ERT profiles.





All other profiles in our study, that neither are situated on an active part of a periglacial landform nor contain ice-rich

material, show an almost unchanged topography and a significant reduction of resistivity values over the last decade. This is particularly true for the Glatzbach catchment, where the topography remained practically unchanged. Here, no significant movement rates are expected due to the absence of furrows and ridges indicating creeping permafrost and the generally lower ice content. However, the profiles in the Glatzbach catchment was re-measured significantly later during the course of the year (early September instead of early July), and observed resistivity changes also partly reflect seasonal differences in

temperature, humidity, or snow cover, (compare Pogliotti et al., 2015; Scherler et al., 2013) as well as the seasonal advance of the thawing front (Hilbich et al., 2008), which results in decreasing resistivity values within the active layer (64–96 m, Figure 11, b).

In the Glatzbach catchment we focussed on the description of the resistivity changes below the active layer (< 2 m, given by Otto et al., 2012) without an influence by seasonally varying conditions. The particular (almost identical in shape) resistivity

distribution along the historical and the repeated profile GL-P2 in (Figure 11, b) supports adequate profile recovery. Resistivity values of up to 3 kΩm below the lower part (0–52 m) of the profile can be linked to a fine-grained, phyllitic substrate outside of the coarse-grained and ice-containing block field (Otto et al., 2012). The regolith and the underlying bedrock are highly conductive due to the lithology and additional tectonic fracturing, as reflected by low resistivity values <1 kΩm. The range and distribution of low resistivity values is in the same range and distribution away from the block field in

the repeated, non-permafrost related part of profile GL-P2.

## 5.3 Permafrost degradation

Documenting permafrost degradation by the comparison of historical and repeated ERT tomograms is based on three major processes: (1) The general atmospheric air temperature increase (MAAT) results in warmer subsurface temperatures (MAGT) in permafrost related environments (Etzelmüller et al., 2020). Subsequently, (2) this leads to a thickening of the

active layer (Mewes et al., 2017; Mollaret et al., 2019), and (3) results in ice-poor permafrost responding stronger to temperature warming than ice-rich permafrost (Haberkorn et al., 2021), as more latent heat is required to thaw permafrost with high ice content than with low ice content (Harris et al., 2009).

We interpret the observed resistivity decrease over time as a proxy for permafrost degradation due to air temperature increase. The closer region of our Swiss study sites hosts long-term air temperature monitoring sites: During the time period

2003 to 2018, the air temperature increased by 1.1 °C (MAAT = 0.07 °C a$^{-1}$) at the Murtèl rock glacier (within the same Corvatsch catchment), by 0.9 °C (MAAT = 0.06 °C a$^{-1}$) at the Schilthorn (Berner Oberland) and by 0.6 °C (MAAT = 0.04 °C a$^{-1}$) at the Stockhorn (Walliser Alpen) (Hoelzle et al., 2021). Weather stations around the Austrian study site follow a similar or slightly stronger trend for the same time period 2003 to 2018: temperature data indicates an increase of 1.2 °C (MAAT = 0.08 °C a$^{-1}$) at Hoher Sonnblick (3106 m a.s.l.) and of 1.1 °C (MAAT = 0.07 °C a$^{-1}$) at Rudolfshütte (2315 m

a.s.l.) (ZAMG, 2021).

For warm permafrost with temperatures close to 0°, which is the case at many permafrost sites in the European Alps, the average warming of the MAGT is usually below 0.03 °C a$^{-1}$ (Smith et al., 2022). The MAGT derived from borehole measurements at the Murtèl rock glacier, which is a comparatively cold permafrost site (PERMOS, 2021), shows an average temperature increase of slightly more than 0.03 °C a$^{-1}$ at 10 m depth during the last 30 years (PERMOS, 2021). No MAGT

data exists in the closer environment of the Austrian study area. However, we also expect an increase in MAGT as a consequence of the air temperature warming.

Our study shows a stronger ice degradation of ice-poor permafrost compared to ice-rich permafrost. We attribute this observation to a faster reaction of ice-poor permafrost thawing to warmer temperatures. All ERT profiles containing ice-poor permafrost tomograms display a significant reduction of resistivity in the repeated tomograms: The resistivity change in the

Corvatsch catchment shows a decrease of more than 10 kΩm, to a current maximum of 30 kΩm. The Austrian Glatzbach catchment follows the same trend: The former maximum of about 80 kΩm is reduced to a present maximum of about 30 kΩm along both repeated ERT-profiles within 12 years (Figure 11). We attribute the resistivity decrease in both catchments to the possibly stronger regional warming (cf. section 5.3.1). Subsurface characterized by ice-poor permafrost of the Gianda Grischa rock glacier follows the same trend of a strong decrease in resistivity (Figure 7 & Figure 8).

At the same time, the ice-rich permafrost at the Gianda Grischa rock glacier partly shows a resistivity increase. Besides the increase of resistivity along profile GG-P2 and GG-P3 (Figure 9 and Figure 10), as discussed in section 5.2, the resistive distribution of ice-rich permafrost remains almost identical of both the historical and the repeated profile GG-P2 (Figure 9). With this finding along profile GG-P2 we see a similar trend as Haberkorn et al. (2021), i.e. that ice-poor permafrost is affected by a more pronounced resistivity decrease than ice-rich permafrost. However, we emphasize that further

comparisons between historical and repeated profiles regarding ice-rich permafrost are needed to support a more general statement.

Additionally, the observed creeping acceleration of the Gianda Grischa rock glacier also indicates a degradation of the permafrost by potentially increased water supply (Eriksen et al., 2018) following general warming (Marcer et al., 2021). However, in spite of the observed resistivity increase in the compression zone, we observe a stronger thawing of the ice-

related permafrost in other areas of the Gianda Grischa (compare Figure 7 & Figure 8). Especially ice-poor permafrost melts faster and leads to a higher water availability and an increase of the ice velocity.

## 6 Conclusion and future work

Our study reveals permafrost degradation by repeating historical ERT measurements on different periglacial features in two regions of the European Alps. The repetition of the profiles was carried out after a period of 10, 16 and 12 years on a

protalus rock glacier, a rock glacier and on a periglacial talus slope. The comparison of the historical ERT-profiles with repeated surveys includes (a) a data enquiry (historical ERT-data, GPS-coordinates, historical photographs, profile characteristics), (b) a proper recovery of the historical profile location for an optimal match of both tomograms for a visual



comparison, (c) an identical data processing of the historical and the repeated ERT-data, (d) the discussion of the resistivity and topography changes in their respective geomorphological and climate-change context.

Our findings reveal an extensive degradation of ice-poor permafrost at all study sites. As indicated by a significant decrease in resistivity, the active layer at all sites thickened and ice-poor permafrost degraded over the course of 10 to 16 years of observation. Surprisingly, ice-rich permafrost of the active rock glacier displayed no resistivity decrease but shows a significant resistivity increase. We suggest that a steep active and accelerating section above the profile location supplies an increasing amount ice and debris to the shallower profile position. Additionally, the average annual creep rate of this section

fastened in the period 2005-2019 compared to 1978-1998, supporting the notion of permafrost degradation in reaction to rising temperatures.

Repeating ERT profiles to detect permafrost degradation is challenged by accurate recovery, by different data acquisition (measurement equipment, electrode array, spacing, profile length) and by geomorphological circumstances that need to be considered for the comparative interpretation of the resistivity data. The application of textile electrodes leads to a significant

reduction of time and effort for the deployment of the electrodes and thus provides important advantages on coarse and blocky surfaces in high mountain environments. Our tests provide convincing evidence that the contact resistance and the related data quality are at least comparable, if not better, to that of conventional steel pikes. However, insufficient boulder coupling in the subsurface can obviously not be compensated, irrespective of the electrode design. The repeated remote sensing approach tracking the local rock glacier kinematics supplements the ERT-based findings of permafrost degradation

by observed velocity increase. However, the use of InSAR data (e.g. Reinosch et al., 2020) provides additional information about vertical changes, such as uplift and subsidence rates. Therefore, we recommend the combination of geophysical data and InSAR time series analyses for future studies of permafrost degradation to reveal volumetric changes of ice-containing permafrost. We conclude that the repetition of historical ERT campaigns combined with remote sensing is a useful approach to decipher the impact of atmospheric warming on permafrost affected landforms in the European Alps.


*Data availability:* The used data set is available for download at https://doi.org/10.5281/zenodo.7348526. Additional metadata can be received by the IDGSP database (https://www.unifr.ch/geo/cryosphere/en/projects/permafrost-monitoring-and-dynamics/idgsp.html) via ertdb@unifr.ch".

*Team list and Author contributions.*

JB applied for funding, designed the study, conducted fieldwork, processed and interpreted ERT data, wrote the manuscript, and conceptualized figures. RG participated in fieldwork, collected temperature data and the information about the study areas and contributed to the discussion. JB, JM, AH and MB developed and designed the textile electrodes, contributed to the discussion. JM carried out the quantitative comparison of different electrode designs. CHa, CHi, CK and RF provided historical ERT data as well as contributed to the discussion. SB and JHB processed remote sensing data and contributed to

the discussion. All authors revised the manuscript carefully.



## Acknowledgements

We thank all colleagues who contributed to this study, especially Robin Zywczok, Quirin Gloor and Emanuel Berchtold for their enormous working effort. We thank Wolfgang Heinz from the mountain hut "Glorer Hütte" and the Corvatsch AG Station Surlej for their support with the equipment transport. This study based on the TransTip StartUp funding, provided the

International Research Training Group (GRK 2309/1) "Geo-ecosystems in transition on the Tibetan Plateau (TransTiP)", funded by Deutsche Forschungsgemeinschaft (DFG).

Work by C. Hauck and C. Hilbich was conducted within the GCOS-Switzerland financed REP-ERT project "Permafrost monitoring by reprocessing and repeating historical geophysical measurements". Additional funding for field work was provided by the Action Group "Towards an International Database of Geoelectrical Surveys on Permafrost (IDGSP)" of the

International Permafrost Association (IPA).

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
