# Peer review of "Identifying mountain permafrost degradation by repeating historical ERT-measurements"

_The Cryosphere, 2022_

## Community Comment (CC1)

[supplement omitted: unrelated document]

---

## Author Comment (AC2)

**General comment: In terms of a general comment, the authors must be careful that permafrost degradation is not limited to changes in ground temperature, but also changes in unfrozen water content, which isn't mentioned once as a process. Latent heat, which is only mentioned once, plays a key role and the changes in resistivity must be evaluated in the context of phase change and not just temperature change. In addition, it would be helpful if the authors provided more information regarding the limitation, accuracy and errors of the various measurements. In other words, when can once trust that a change is a real change and not related to the limitation of the measurements.**

We agree and discuss the issue in a new section of chapter 5.4 (compare comment to Line 510 and 525). We add in Line 54: *"However, the thawing and freezing processes around the freezing point can be delayed due to latent heat fluxes associated with phase change (ice–water), rather than warming or cooling of the subsurface (Mollaret et al., 2019)."* and in Line 194 *"latent heat fluxes"*. In addition, we focused our interpretation of resistivity values on changes greater than 10 kΩm to avoid an interpretation that includes smaller resistivity changes as an effect of (seasonal) cooling/thawing. This automatically implies an exclusion of the interpretation of the change in thickness of the active layer as an indication of permafrost degradation. To clarify this, we add in Line 206: *"However, we limit our interpretation to resistivity values and resistivity changes exceeding 10 kΩm as fast changing small-scale and seasonal effects (e.g. unfrozen water content, compare Fortier et al., 1994) induce resistivity changes that cannot be distinguished from resistivity changes associated with perennial external temperature changes. This also excludes a detailed interpretation of an active layer-thickness increase as an indication of permafrost-body thaw."*

Information regarding the limitation, accuracy and errors is given in a new chapter called *"5.2 Tomogram comparability and accuracy assessment"*

**Line 16: Delete "mountain" as the statement is true for all of the permafrost around the world**

We have deleted "mountain".

**Line 20: "Alpine". Please try to be consistent throughout the manuscript. Alpine, with a capital A, reference to the Alps, whereas alpine with a small a references mountain environments with alpine climate, but are geographically not restricted to the Alps. Finally, and the most general term is "mountain". In the context of the paper I suggest to use mountain permafrost consistency.**

We agree and revised/rephrased the manuscript (Line 20, 40, 41, 42, 50, 62, 78, 427, and 450) regarding the use of "mountain" instead of "alpine".

**Line 24: Delete "periods of", redundant wording.**

We agree and have deleted "periods of".

**Line 28: Creep instead of Creeping (check also other places in the manuscript!)**

We agree and revised/rephrased the manuscript (Line 28, 101, 118, 456, 457, 476, 492, and 542) regarding the use of "creep" instead of "creeping".

**Line 35: "mountain" instead of "mountainous"**

We changed "mountainous" to "mountain".

**Line 37: Add Arenson et al. (2022) as reference:**

Arenson, L. U., Harrington, J. S., Koenig, C. E. M., & Wainstein, P. A. (2022). Mountain Permafrost Hydrology—A Practical Review Following Studies from the Andes. *Geosciences*, *12*(2), 48. https://doi.org/10.3390/geosciences12020048

We added the reference.

**Line 44: "used to distinguish ice-poor from ice-rich" replace with "used to differentiate between ice-poor and ice-rich"**

It is replaced.

**Line 48: delete "of"**

"of" is deleted

**Line 48: what about RST?**

Line 49 is changed accordingly: *"As the installation of boreholes is cost and time intensive, indirect methods such as electrical resistivity tomography (ERT), seismic refraction tomography (SRT), ground penetrating radar (GPR) are often applied for subsurface ice detection and permafrost characterisation (Halla et al., 2021; Maurer and Hauck, 2007; Vonder Mühll et al., 2002)"*

**Line 69: mountain permafrost, not mountainous permafrost**

We agree and revised the manuscript accordingly.

**Line 112: Following the latest guidelines from the IPA Rock glacier inventories and kinematics working group, I suggest to call this a talus derived rock glacier and not a protalus rock glacier.**

We agree and changed Line 99, 112, 118, 123, 263, 304, 317, 550.

**Line 125: When using a name, rock glacier, is typically capitalize (this applies at various locations in the manuscript), i.e. Gianda Grischa Rock Glacier**

We changed Line 100, 107, 111, 113, 128, 129, 142, 225, 269, 326, 334, 343, 360, 376, 389, 433, 460, 469, 515, 523, 534, 535, and 542.

**Line 135: Delete "in this study"**

It is deleted.

**Line 153: Delete "by -1.3 °C"**

It is deleted.

**Line 156: "Otto et al. (2012)"**

We added the full stop.

**Line 200ff: but also, as permafrost degrades and the amount of unfrozen water increases. Even small amounts of unfrozen water can have significant effects on the electrical resistivity.**

Please refer to the main comment and the resulting changes.

**Line 204: delete "perennial"**

Is deleted

**Line 224: "large" instead of "huge"**

Is replaced

**Line 255: "delete "ice" before maximized.**

We rephrased the sentence as follows: *"...onto the ground maximizing the contact area."*

**Line 300: " had any lab testing being performed with these electrodes, to evaluate the effect under controlled conditions? I find it very difficult to evaluate the effect if the conditions are not properly controlled. I.e. only a very general and qualitative conclusion is possible at this point.**

This point was also raised by Reviewer 1. We relied on earlier investigations of the textile electrodes within a bachelor's thesis (see Westphal et al. 2022: (https://dgg2022.dgg-tagung.de/englisch/conference-booklet/)), in which the contact impedances of textile electrodes were investigated and compared with conventional steel electrodes at different surfaces. The main conclusion is that the textile electrodes perform as well as steel electrodes as long as a small amount of water is used to moisten the textile. Following these guidelines, we had no concerns about contact impedances during this survey. We added a few sentences in section 3.3 summarizing the results of previous investigations, including actual values of contact impedances: We added in Ln 271: *"Our design results in an approximately circular contact area with a diameter of roughly 15 cm. Contact resistances were investigated prior to the campaign*

*over different surfaces and compared with those of conventional steel electrodes. It was found that the textile electrodes generally perform as well as steel electrodes as long as a small amount of water (e.g. 80 ml) is being used to moisten the textile. For example, on a semi-paved surface, both electrode types provided resistances between 2 and 5 kΩ, and even on a hard gravelly surface where the steel electrodes could not be used, the average contact resistance of the textile electrodes was below 6 kΩ, which is far below the threshold above which the equipment cannot reliably measure any more (> 1 MΩ for the Geotom)."*

**Line 310: Delete "nevertheless"**

Is deleted.

**ERT tomograms. Please add the following parameters from the overview tables (e.g. Table 1) to all the tomograms: Electrode Spacing, No. of Iterations and RMS error [%].**

The parameters were added.

**Line 316: But also increase in unfrozen water content. Decreasing resistivity is a combination of two processes.**

We added in Line 317 *"...and probably an increase in moisture."*

**Line 317: this could be a result of seasonal conditions and not necessarily the result of long term change, i.e. wetter summer**

The sentence was deleted according to the main comment.

**Figure 6 (and other figures): Make sure that the y-Axis are all the same**

We changed the y-Axes to be all the same

**Figure 6 (and other figures): Make sure that EVERY number has a unit, or that the units are clearly indicated. Distances are often not labeled with the appropriate unit.**

We added in Fig.6-Fig.11 by *"Elevation [m.]"* and "*Distance [m.]*"

**Line 367 (and other places); Permafrost doesn't melt, because permafrost is a thermal state. Use degrade in the context of permafrost. If you specifically refer o the ground ice, then you can use melt, but ice must be indicated.**

We replaced melt by *"degrade"* in Line 367 and 546.

**Line 370: 16-year period**

We changed the phrase.

**Line 374 (and other places): Make sure to not use massive ice and ice-rich permafrost as synonyms. I suggest to use ice-rich permafrost in this paper, unless you have physical evidence, e.g. from drilling, that there is massive ice. (e.g. also Line 399).**

We agree and changed Line 204, 353, 374, 383, 399, 461.

**Line 395 ff: You have to be careful that increase in resistivity is not automatically attributed to poor data while decrease in resistivity to permafrost degradation. The interpretation of your data cannot be biased towards the second.**

We agree that this bias should be avoided. Since this section is not essential for the main message, we deleted Line 395, 396, 397.

**Line 414 (but also other instances): Be careful with preposition "of" and "in". In this case "in" would be better. When using "reduction of" the focus on is the reduction itself, but when using "in" the focus is on what is reduced. In this case, the focus is the resistivity, because that is what you are measuring and comparing. On line 417 you have a similar situation with "similarity" or in line 472 with "increase".**

We replaced "of" by "in" in Line 414, 417 and 472.

**Line 427: This isn't just rue for alpine settings**

Yes, therefore we deleted "in alpine settings"

**Line 444: delete "of"**

Is deleted

**Line 456: creep instead of creeping**

The entire manuscript was checked and creeping was replaced by *"creep"*.

**Figure 12b: arrow lengths should be scaled according to velocity**

The arrow length was adapted.

**Line 510: "stronger" is relative. You are just limiting the response to temperature. Changes in unfrozen water content w/o major change in temperature may also be labeled as a strong response.**

**Line 525: "increase in MAGT": Don't just focus on temperatures. The manuscript provides the impression as if permafrost degradation is equal to warming. This is incorrect as changing state, i.e. the thawing of ground ice, is very critical. Changes in resistivity are perfect to record such changes. This must be discussed in more detail and the discussion should not be limited to changes in MAGT.**

We agree with both comments (L510 and L525). We have to state out, that effect of unfrozen water content influence significantly the change in resistivity values. We changed the title of the chapter in *"Permafrost degradation in terms of resistivity change and temperature increase"* and introduced the chapter as follows:

*"Resistivity values reflect different subsurface conditions (c.f. chapter 3.1), and are especially sensitive to the occurrence of ground ice. However, unfrozen water content, infiltration of precipitation as well as snow and ice melt lower resistivity values, even if the subsurface temperature is below the freezing point (Hilbich et al., 2011; Kneisel et al., 2008; Mollaret et al., 2019). Unfrozen water content is available between 0°C and -5°C depending on the material characteristics and lowers resistivity up to 600 $\Omega$m for sandy soil in a laboratory experiment (Tang et al., 2020). The effect of unfrozen water content in the investigated study sites should occur at higher resistivity values due to the larger grain sizes and the resistive lithology. Fortier et al. (1994) examined the effect of unfrozen water content on apparent resistivity during active layer thickening (April – June). The unfrozen water content significantly affects the apparent resistivity up to a value of about 10 k$\Omega$m, while the ice content increases above this value: In the frozen layer characterized by high content of ice (mass proportion of ice > 30%) below the active layer, the unfrozen water content decreases significantly and the apparent resistivity exceed the value of around 10 k$\Omega$m. Therefore, we assess the effect of unfrozen water content on the interpretation of permafrost degradation in this study as minor due to used threshold (10 k$\Omega$m). Additionally, values around 10 k$\Omega$m indicate the presence of ice also in other studies (c.f. Hauck and Kneisel, 2008; Otto et al., 2012) based on different subsurface materials than in the study of Fortier et al. (1994). A delay in resistivity change can be observed during thawing/freezing due to the effect of latent heat around 0°C, the so called zero curtain effect (Farzamian et al., 2020; Mollaret et al., 2019). In general, resistivity-temperature relationships show exponential behaviour below the freezing point (Hauck, 2002). Furthermore, ice-poor permafrost is more sensitive to temperature warming than ice-rich permafrost (Haberkorn et al., 2021), as more latent heat is required to thaw permafrost with high ice content than with low ice content (Harris et al., 2009)."*

**Line 528: "faster reaction" ignores the effect of latent heat, which is very energy intensive. But is also a fast reaction, it just doesn't manifest itself so quickly; and is much more difficult to measure and observe in the field.**

The sentence was deleted.

**Line 545: degrade not melt**

Melt is replaced by degrade.

**Section 5, Discussion: I would have liked a discussion on the errors / accuracy of the measurements and inversion techniques used. How much of the changes noted may be**

**attributed to errors and other uncertainties in the measurements? What is the level of change at which one can confidentially say that the properties of the ground have really changed?**

We absolutely agree and add a corresponding chapter called *"5.2 Tomogram comparability and accuracy assessment"*. By introducing the study of (Fortier et al., 1994) we discuss in chapter 5.4 the value of 10 kΩm as a reasonable threshold for a confidentially interpretation of permafrost degradation by resistivity changes. ERT as an indirect method does not achieve absolute reliability.

*"5.2 Tomogram comparability and accuracy assessment*

*The exact re-location of the historical profiles, different measurement equipment, electrode array, spacing as well as profile length make a comparison of historical and repeated tomograms challenging, resulting in a different data resolution and in a different depth of investigation indicated by the varying black tomogram outlines (c.f. Figures 6 - 11). Therefore, some constraints must be taken into account when comparing the tomograms. For instance, we are limited to a visual comparison based on significant anomalies greater than 10 kΩm in both historical and repeated tomograms. By using this threshold, we suppose that most of the uncertainties (proper location, unfrozen water content, water chemistry and saturation, pore connectivity, mineralogy and grain size characteristics) can be excluded (compare Fortier et al., 1994 and Mollaret et al., 2019). A direct, algorithm-based comparison seems to be an important and challenging task for the future. Temperature related changes in active layer thickness, which is strongly sensitive to climate warming (Scherler et al., 2013) cannot be inferred reliably due to the limitation of an visual comparison.*

*Regarding the tomogram comparability, we used the smoothness-constrained least-squares inversion (L2-Norm) instead of the often applied robust inversion scheme (L1-norm) (Emmert and Kneisel, 2017; Halla et al., 2021; Supper et al., 2014, etc.). This way, we avoid the tomograms to be dominated by sharp boundaries, the comparison of which would be misleading due to the location errors discussed above. The resulting RMS errors vary in a range from 3.2 % to 16.3 %, whereby the RMS errors are slightly higher in the tomograms from the Gianda Grischa rock glacier. Yet these values are well within the range of other ERT applications on rock glaciers (Hauck and Kneisel, 2008; Hauck and Vonder Mühll, 2003; Villarroel et al., 2021). The number of iterations needed to fit the measured apparent resistivity to the calculated resistivity model counts 4 +/-1, which is in a similar range of values of the mentioned, comparable studies. Another aspect of data quality concerns the number of filtered values compared to all collected data points. The raw data of the Gianda Grischa rock glacier were filtered by 0-20% of all data points, whereby only 0-6.1% from the raw data were filtered at the Corvatsch catchment, and 0-8.7% at the Glatzbach catchment. A similar filter approach as we used (compare section 3.1) is presented by Rosset et al. (2013) called technical and magnitude filter. The authors filtered an average of 8,79% of all data points in six tomograms of four comparable study sites. The similar filtering approach used for the 16 tomograms of this study results in an average of 6.8% filtered values."*

Line 559: amount "of" ice

"of" is added

Line 560: "increased" instead of "fastened". Note: a rate cannot get faster. Only a velocity can

"increased" is replaced

Line 562: "…, by different data acquisition(measurement equipment, electrode array, spacing, profile length) and by geomorphological circumstances that need to be considered for the comparative interpretation of the resistivity data." This statement only appears in the conclusions, but this should be discussed further in the previous sections. It is very important and essential for the completion of similar studies in the future.

We agree and can now refer on the new chapter 5.2, where we included the sentence *"A direct, algorithm-based comparison seems to be a very important but challenging task for the future."* Additionaly, we changed the title of the chapter 5.3: *"Geomorphological interpretation of resistivity change"*. In our opinion, this statement is now substantiated in the chapters 5.2 and 5.3.

Additionally, we added in Line 564: *"A useful future task will be to develop an algorithm-based approach that goes beyond a visual comparison of the historical and the repeated ERT tomograms."*

Line 568: Delete "obviously"

Is deleted

References:

[revised manuscript text omitted]

---

## Author Response (AR2)

Dear Prof. Dr. Jacopo Boaga!

I hope you started well into the new year. Thank you very much for your review and your encouraging as well as important comments. This letter contains point by point responses on your comments. New/rephrased sentences are indicated by italic letters.

Ln 55-60 This paragraph seems to introduce ERT use in permafrost studies, but most of the references here cited are not relative to ERT application case studies. It sounds strange.

We carefully scrutinized citations to permafrost studies regarding typical periglacial landforms (protalus ramparts, talus slopes and rock glaciers). All the mentioned studies detected and characterized subsurface ice by using ERT except one: The citation "Kenner, R., Phillips, M., Beutel, J., Hiller, M., Limpach, P., Pointner, E. and Volken, M.: Factors controlling velocity variations at short-term, seasonal and multiyear time scales, Ritigraben rock glacier, Western Swiss Alps, Permafr. Periglac. Process., 28(4), 675–684, doi:10.1002/ppp.1953, 2017." was replaced by the correct citation *"Kenner, R., Phillips, M., Hauck, C., Hilbich, C., Mulsow, C., Bühler, Y., Stoffel, A. and Buchroithner, M.: New insights on permafrost genesis and conservation in talus slopes based on observations at Flüelapass, Eastern Switzerland, Geomorphology, 290(April), 101–113, doi:10.1016/j.geomorph.2017.04.011, 2017."*

The sentence was re-phrased to: *"ERT measurements have also been successfully applied to detect and characterize permafrost on periglacial landforms as…"*

Ln 66-70 I'm not sure statistical approach can solve the problems of contact resistance, maybe this sentence should be re-phrased.

We re-phrased the sentence as follows: *Emerging challenges such as changing contact resistances, different instruments or inversion artefacts can be addressed, for example, by statistical analysis (Supper et al., 2014) or adapted data processing schemes (Oldenborger and LeBlanc, 2018).*

Ln 85  Here or in line 440, for specific galvanic contact in debris problem consider also https://doi.org/10.1002/nsg.12192

We agree and have now included the mentioned publication in Ln 85 and in the citation list.

Ln 179-186 No clear the relative standard deviation 0.3%. Is this the stack error? Authors for sure appreciated how pre-processing is essential in ERT inversion. The filtering of the dataset should

be then better described, if it is done with some pre-processing code or in the RES2DINV suite. 'Visually identifiable' is a weak approach for repeatable measurements.

This unclear paragraph was re-written as follows:

*"An initial quality assessment of the field data was achieved during the data acquisition 2021 by using the GeoTom software (V. 7.19, Geolog 2000). In a first step, the data quality of each quadrupole measurement is checked visually by comparing its value with neighbouring points in the pseudo section. Large deviations or outliers (e.g. given by a defect in one cable which produced random values for measurements with one specific electrode) indicate poor data quality. In a second step, (a) all data points with a relative standard deviation of all stacks (stack error) above 0.3 %, and (b) all data points collected with the maximum value of the input voltage, which indicates that no sufficient signal strength can be achieved, were identified. Visually identified outliers as well as the values identified by (a) and (b) were re-measured after checking the placement of the electrodes and improving the contact between the electrode and the surface by adding salted water. Remeasured data points still meeting one of the criteria (a) or (b) or being obvious outliers were deleted and reconstructed by interpolating the neighbouring values before the inversion. Because information on stack error and input voltage was unavailable, this process could not be performed on the historical data. Instead, the historical data were filtered manually using the "Exterminate bad datum points"-function in the RES2DINV software. Bad data points are values with apparent resistivity that are apparently too large or too small compared to the neighbouring values. The overall number of filtered data points is given in tables 2 to 4 for the three surveyed catchments."*

Which error was used during the inversion process?

The inversion in Res2DInv was carried out without taking information on the data error into account.

Ln 252-260 The very important point highlighted by Uhlemann and Kuras should be inserted here, since breaking the point assumption is the first doubt arising from this (very interesting) textile approach.

In the corresponding paragraph in the Methods and Data section, we added information (Ln 258) on the actual electrode size ("*Our design results in an approximately circular contact area with a diameter of roughly 15 cm.*") and make reference to the discussion in 5.1 ("*Possible effects of the large electrode size and the violation of the point-source assumption during the inversion of the ERT data will be addressed in the discussion section.*").

The discussion of possible effects of the electrode size in section 5.1 is further extended as follows:

*"For capacitively coupled electrodes, Uhlemann and Kuras (2014) argue that the point-pole approximation is valid, if the electrode spacing is at least 4 times the diameter of the contact area. Since the diameter of the contact area of our new textile electrode is <15 cm and our minimum electrode spacing is 4 m, we are well within these limits. To assess the effect of the size of square surface electrodes on ERT measurements, Cardarelli and De Donno (2019) carry out finite-element modelling studies. They find that for electrode separations five times the electrode size or larger, the relative error compared to a point source falls well below 1 %. This is in good agreement with the findings by Uhlemann and Kuras (2014) and further supports the feasibility of our relatively small textile electrodes."*

Ln 278 Input voltage ? Do you mean current injection ?

Changed to *"...the input voltage used for current injection."*

Ln 285-290 This is my main criticism to the work: authors do not present a quantitative comparison of the most relevant aspect of these new electrodes: contact resistance. I expect that, as first testing of these very interesting approach, authors measure and compare contact resistance in KOhm. Did you measure contact resistance before collecting measurements ? Which range was measured? Did you compare textile and electrode contact resistance during the hybrid line collected ? When different instruments were used, internal resistance problem should be addressed too in the comparison of the electrodes performance.

In fact we did not measure contact resistance at this particular site, because under these difficult logistical conditions we tried to be as time-efficient as possible. We relied on earlier investigations of the textile electrodes within a bachelor's thesis (see Westphal et al. 2022: (https://dgg2022.dgg-tagung.de/englisch/conference-booklet/)), in which the contact impedances of textile electrodes were investigated and compared with conventional steel electrodes over different surfaces. The main conclusion is that the textile electrodes perform as well as steel electrodes as long as a small amount of water is used to moisten the textile. Following these guidelines, we had no concerns about contact impedances during this survey. We added a few sentences in section 3.3 summarizing the results of previous investigations, including actual values of contact impedances.

Moreover, in order not to exceed the scope of this manuscript, a compromise had to be found to present different, very interesting aspects (permafrost degradation, geomorphological interpretation, remote sensing, ERT) in this manuscript. We focused on permafrost degradation and the geomorphological interpretation by ERT and remote sensing data. The use of the textile electrodes is (not mentioned in the title) therefore helpful, but detailed analyses can be expected in a separate manuscript in order not to go beyond the scope of the present manuscript here.

We added in Ln 271: *"Our design results in an approximately circular contact area with a diameter of roughly 15 cm. Contacted resistances were investigated prior to the campaign over*

*different surfaces and compared with those of conventional steel electrodes. It was found that the textile electrodes generally perform as well as steel electrodes as long as a small amount of water (e.g. 80 ml) is being used to moisten the textile. For example, on a semi-paved surface, both electrode types provided resistances between 2 and 5 kΩ, and even on a hard gravelly path where the steel electrodes could not be used, the average resistance of the textile electrodes was 6 kΩ, which is far below the threshold above which the equipment cannot reliably measure any more (> 1 MΩ for the Geotom).*

*During the survey described here, the surfaces were generally rugged, blocky and not flat, and therefore additional measures had to be taken. We placed wet sediment and sediment containing biotic material (e.g. roots, moss) between the textile electrode and the rugged surface of boulders."*

Ln 395 Here contact resistance is cited but without values, as in ln 567.

See comment above. Unfortunately, no information on the actual contact resistances during the data collection in the field is available.

Ln 558-560 This speculation about increasing in resistivity is very interesting and maybe need more space in the discussion, rather than in the conclusion. Here again injected current and contact resistance may play a relevant role and should be compared in the time-repeated ERT section.

The increasing resistivity during the period of 16 years is indicated in two profiles (GG-P2 and GG-P3). We attribute this resistivity increase to a geomorphic background and not to a technical issue. If this increase should be attributed to contact resistances or the potential injected current, it would also have to be recognizable in other profiles (e.g. GG-P1 and GG-P4).

Furthermore, as long as the contact resistance of the textile electrodes is not orders of magnitudes higher than the contact resistance of traditional spike electrodes, we do not expect any significant effect of the contact resistance on the reconstructed resistivity values in the subsurface. This can be seen in Figure 6 of the manuscript, which compares resistivity models reconstructed from three different electrode-substrate coupling types (steel pikes, steel pikes with sponges, textile electrodes). As long as a reasonable contact can be established between the substrate and the electrode, the apparent resistivity measured with a four-point setup should, in principle, be independent of the contact resistance.

In all all the ERT sections I suggest to increase fonts of axes, legend scale and labels.

We agree and all fonts were increased.

Best regards, Johannes Buckel

References:

Cardarelli, E. and De Donno, G.: Chapter 2 - Advances in electric resistivity tomography: Theory and case studies, in Innovation in Near-Surface Geophysics, edited by R. Persico, S. Piro, and N. Linford, pp. 23–57, Elsevier., 2019.

Westphal K., Mudler J., Buckel J., Bücker M., Hördt A. (2022): Die Anwendung von Textilelektroden bei geoelektrischen Widerstandsmessungen. 82. Jahrestagung Deutsche Geophysikalische Gesellschaft,07.–10. März 2022, München (online) (Abstract&Poster)

---

## Author Response (AR3)

Dear Prof. Schweizer!

We would like to thank you very much for your editing work. We added the citation Pavoni et al. 2022 in Ln 85 and in the references. Additionally, we carefully read the manuscript again and corrected some minor issues due to missing homogenity (double spaces, no spaces, missing spaces between the number and the unit). We allow us to insert an additional citation (Hartl et al. 2023) in the section 5.3 due to its novelty and importance for the discussion. Additionally, we replaced in Ln 174 "good" by optimal and in Ln 175 "good" by high.

All the best, Johannes Buckel

Hartl, L., Zieher, T., Bremer, M., Stocker-Waldhuber, M., Zahs, V., Höfle, B., Klug, C., and Cicoira, A.: Multi-sensor monitoring and data integration reveal cyclical destabilization of the Äußeres Hochebenkar rock glacier, Earth Surf. Dynam., 11, 117–147, https://doi.org/10.5194/esurf-11-117-2023, 2023